# Links between Neuroanatomy and Neurophysiology with Turning Performance in People with Multiple Sclerosis

**DOI:** 10.3390/s23177629

**Published:** 2023-09-03

**Authors:** Clayton W. Swanson, Brett W. Fling

**Affiliations:** 1Brain Rehabilitation Research Center, Malcom Randall VA Medical Center, Gainesville, FL 32608, USA; clayton.swanson@ufl.edu; 2Department of Neurology, University of Florida, Gainesville, FL 32608, USA; 3Department of Health and Exercise Science, Colorado State University, Fort Collins, CO 80521, USA; 4Molecular, Cellular, and Integrative Neuroscience Program, Colorado State University, Fort Collins, CO 80521, USA

**Keywords:** multiple sclerosis, turning, motor cortex, TMS, MRI, inhibition

## Abstract

Multiple sclerosis is accompanied by decreased mobility and various adaptations affecting neural structure and function. Therefore, the purpose of this project was to understand how motor cortex thickness and corticospinal excitation and inhibition contribute to turning performance in healthy controls and people with multiple sclerosis. In total, 49 participants (23 controls, 26 multiple sclerosis) were included in the final analysis of this study. All participants were instructed to complete a series of turns while wearing wireless inertial sensors. Motor cortex gray matter thickness was measured via magnetic resonance imaging. Corticospinal excitation and inhibition were assessed via transcranial magnetic stimulation and electromyography place on the tibialis anterior muscles bilaterally. People with multiple sclerosis demonstrated reduced turning performance for a variety of turning variables. Further, we observed significant cortical thinning of the motor cortex in the multiple sclerosis group. People with multiple sclerosis demonstrated no significant reductions in excitatory neurotransmission, whereas a reduction in inhibitory activity was observed. Significant correlations were primarily observed in the multiple sclerosis group, demonstrating lateralization to the left hemisphere. The results showed that both cortical thickness and inhibitory activity were associated with turning performance in people with multiple sclerosis and may indicate that people with multiple sclerosis rely on different neural resources to perform dynamic movements typically associated with fall risk.

## 1. Introduction

Multiple sclerosis is a chronic, immune-mediated, demyelinating neurological disease of the CNS. Among young adults, multiple sclerosis is the major cause of neurological impairment, leading to irreversible long-term disability throughout the disease course [1]. People with multiple sclerosis (PwMS) present with deleterious neural adaptations that contribute to symptom severity and disease progression [2,3,4,5]. These adaptations can lead to irreparable consequences ranging from transient dysfunction to irreversible impairments [6].

Progressive brain atrophy is a well-known feature of multiple sclerosis and considered irreversible brain damage affecting both gray and white matter [7]. Gray matter adaptations arise early and have been reported prior to clinical diagnosis [7,8,9,10,11,12]. Additionally, studies have shown gray matter alterations demonstrate stronger associations with motor and cognitive dysfunction when compared to lesion accumulation [11,13,14]. While neuroanatomical adaptations are relevant hallmarks in multiple sclerosis, neurophysiological modifications impact symptomology.

A common non-invasive brain stimulation technique applied to evaluate motor cortex associated neurophysiological activity is transcranial magnetic stimulation (TMS) [15]. The use of TMS over the motor cortex can provide relative measures of excitatory (i.e., glutamatergic) and inhibitory (i.e., GABAergic) corticospinal activity [16]. In neurotypical adult brains, the homeostatic balance of excitatory and inhibitory neurotransmission ensures proper neuronal functioning, although in multiple sclerosis this balance appears disrupted [17,18]. Though not fully understood, the excitatory imbalance is thought to result from large quantities of glutamate release by activated immune cells during periods of inflammation [19,20]. This imbalance leads to excess release of extracellular glutamate and subsequently an excitotoxic environment for neural tissues [20]. While excitatory levels naturally fluctuate, relapses, lesion formation, and disease progression are associated with periods of increased extracellular glutamate [21,22,23,24]. As such, much of the TMS literature has focused on assessing excitation in PwMS [25,26,27,28,29,30]. Although less common, researchers have used TMS to assess inhibitory activity, which has been associated with cognitive function and motor performance in populations, including multiple sclerosis [17,31,32,33]. Though most motor performance research has focused on assessing the upper limbs, the neural control of the lower limbs has received less attention [15,18,25,26,30,34,35].

Though the neural control of walking remains a topic of continued investigation, there is agreement that walking involves spinal and supraspinal neural control mechanisms [36,37,38]. A growing body of literature detailing supraspinal contributors in response to complex walking tasks has revealed neuronal firing and cortical thickness patterns associated with complex locomotor movements [36,39]. For instance, complex locomotor movements such as turning while walking have been associated with corticospinal neurophysiological function, indicating that levels of corticospinal inhibitory activity relate to turning kinematics in healthy young and older adults [33], implying that supraspinal contributions are inherit and necessary for effective locomotor performance. While neuroimaging studies have examined associations between neuroanatomical structure, neurophysiological function, and walking, they are generally focussed on straight-ahead walking. However, the neural underpinnings associated with turning while walking remain elusive, especially in populations where turning performance is known to be compromised, such as those with multiple sclerosis. 

The present study aimed to assess the neuroanatomical and neurophysiological correlates of turning performance in people diagnosed with relapsing remitting multiple sclerosis (RRMS). We developed three primary hypotheses, first that PwMS would demonstrate reginal thinning within bilateral primary motor cortices compared to neurotypical control participants. Second, we hypothesized that PwMS would demonstrate reduced excitatory and inhibitory corticospinal activity compared to neurotypical controls. Lastly, we hypothesized that PwMS would demonstrate a positive association between motor cortex thickness and turning performance, a positive association between corticospinal excitation and turning performance, and a negative association between corticospinal inhibition and turning performance.

## 2. Materials and Methods

### 2.1. Population

In total, 26 individuals with RRMS and 23 healthy controls completed two separate laboratory visits. Participants were excluded if they were unable to walk or stand for 10 min without the use of an assistive device, having any MRI- or TMS-related contraindications such as non-MRI compatible implanted medical devices, implanted ferrous metal or metal fragments, facial tattoos or permanent makeup, cochlear implants, having a personal or family history of epilepsy, or currently taking medications known to lower seizure threshold or known to be a contraindication for TMS, and/or having any additional musculoskeletal or vestibular condition. In addition, healthy control participants were free from any clinically diagnosed neurological condition or disease known to impact mobility. Further study enrollment details are displayed in Figure 1. This study was approved by the Colorado State University Institutional Review Board (IRB# 18-7738 H), and all participants provided written informed consent prior to participation.

### 2.2. Turning Acquisition

Participants performed three separate 360° turn trials at their self-selected fast pace, and 1-min of continuous but alternating 360° turns at their self-selected natural pace. To encourage natural turning participants were instructed not to spin or conduct a military style turn. All 360° turns were conducted in an open space while barefoot with research staff spotting. Participants also completed a series of 180° turns during a self-selected pace 2-min walk test. Participants were instructed to turn as if forgetting something in a room they had just left.

### 2.3. Turning Processing

Both the 360° and 180° turn metrics were collected using Opal wireless inertial sensors and quantified through previously validated Mobility Lab software (Version 2) (Opal Sensors, APDM Inc., Portland, OR, USA) [40]. The primary turning metrics for 360° in-place fast turns included, turn duration (s), turn angle (°/s), and peak turn velocity (°/s). The primary turning metrics for 1-min 360° in-place turns included turn duration (s), turn angle (°/s), peak turn velocity (°/s), and number of turns completed (#). The primary metrics for 180° turns included turn duration (s), turn angle (°/s), peak turn velocity (°/s), and number of steps in the turn (#).

### 2.4. MRI Acquisition

All participants underwent an MRI protocol on a Siemens 3T MAGNETOM Prismafit equipped with a 32-channel head coil. MRI protocol included: T1-weighted magnetization-prepared rapid gradient-echo (MP-RAGE) (repetition time (TR)/echo time (TE): 2400/2.07 ms, inversion time: 1000 ms, flip angle: 8°, echo train length: 0.49 ms, field-of-view: 256 mm (180 mm (RL), 256 mm (AP), 256 mm (FH)), slices: 224 (sagittal), resolution: 0.8 × 0.8 × 0.8 mm^3^); and a T2-weighted fluid-attenuated inversion recovery (FLAIR) (TR/TE: 6000/428 ms, inversion time: 2000 ms, echo train length: 933 ms, field-of-view: 256 mm (176 mm (RL), 256 mm (AP), 256 mm (FH)), slices: 176 (sagittal), resolution: 1.0 × 1.0 × 1.0 mm^3^).

### 2.5. MRI Processing

Global and regional cortical thickness measures were reconstructed using the FreeSurfer (Version 6.0.0)-recon-all processing pipeline (Figure 2) [41,42,43]. The T1 and T2-FLAIR images underwent the multimodal recon-all processing pipeline known improve cortical parcellations and segmentations [44]. Following quality assurance of each scan, a priori precentral and paracentral gyri regional thickness measures were exported for further analysis. The precentral and paracentral regions were defined using the Desikan-Killiany atlas through FreeSurfer [45]. 

### 2.6. Muscle Strength Acquisition

Participants produced a series of maximal voluntary contractions (MVCs) to determine the maximal force output of each tibialis anterior (TA) muscle. Participants’ legs (individually) were secured to a platform using a strap secured around the dorsum of the foot and around the heel to limit posterior foot translation. The dorsum strap was secured to a high-capacity carabiner and stationary force transducer. Participants performed between two and five MVC trials that were analyzed for maximal force output. MVC trials were separated by 2 min and concluded when force production no longer increased across trials and the two highest force values were within 10% of each other. The same process was replicated for the opposite leg.

### 2.7. TMS Acquisition

Prior to TMS, participants were comfortably seated with both feet placed on the stationary platform. Electromyography (EMG) electrodes were placed on the muscle belly of each TA using bipolar Ag-AgCl surface electrodes. Following EMG placement, TMS was delivered independently over each motor cortex targeting the contralateral TA muscle. Initial ‘hot spot’ locations were defined as being 1 cm posterior and 1 cm lateral to either side of the vertex. Based on each location, the coil was systematically and incrementally moved until the stimulation response on the contralateral TA produced the largest and most consistent MEP response. Single TMS pulses were deliver using a 2 × 95 mm angled butterfly coil (120-degree, Cool D-B80, MagVenture) positioned tangentially against the scalp at roughly 45–65° from the mid-sagittal line [16,46]. The resting motor threshold (RMT) was determined in both hemispheres and defined as the lowest stimulator intensity to elicit an MEP with a peak-to-peak amplitude of ≥50 μV in five out of ten trials. Two 3-min trials were performed, during which time participants were asked to sustain an isometric dorsiflexion at 15% of their MVC while stimulations were delivered every 7–10 s to the corresponding hemispheric ‘hot spot’ at 120% of the RMT, ensuring a corticomotor response. Each participant received a median of 21 stimulations per hemisphere. 

### 2.8. TMS Processing

EMG data were collected at 2000 Hz (BIOPAC Systems, Inc., Santa Barbara, CA, USA). Offline, EMG data were filtered using a combination bandpass filter (5–500 Hz) with a 60 Hz Nyquist filter through AcqKnowledge software. Filtered data were then imported to MATLAB (MathWorks, Nantick, MA, USA) then rectified and processed using a custom MATLAB script used to identify and quantify TMS measures. To analyze TMS measures, 100 ms prior to simulation and 350 ms post-stimulation were extracted from the EMG trace for each stimulation. All EMG trace segments were visually inspected for quality; stimulations not resulting in a clear or expected stimulation response were removed from further analysis. The remaining traces were averaged together and underwent processing to quantify excitatory and inhibitory responses. Excitatory measures via MEPs are suggested to signify corticospinal excitability, largely used as a proxy for assessing relative levels of glutamatergic activity [47]. To assess GABAergic inhibitory activity, three measures embedded within the silent period were assessed. Conventionally, the cortical silent period (cSP) is quantified as the duration in which muscle activity is diminished following the MEP, where shorter durations indicate reduced inhibitory activity. In the current manuscript, we assessed cSP duration as well as the average percent depth of the silent period (%dSP_AVE_), and the maximum percent depth of silent period (%dSP_MAX_) (Figure 3) [46,48].

### 2.9. Statistical Analysis

Statistical analysis was conducted in JMP Pro 15 with alpha levels set to 0.05 unless indicated otherwise. Between-group sex differences were examined using a chi-square test, all other between-group demographic variables were assessed using a two-sample t-test. All data are presented as mean ± SD unless noted otherwise. 

To assess for differences between the three, 360° fast pace turn trials a repeated measures analysis of variance was performed. No significant differences were observed and, therefore, all variables were averaged together. To assess between-group differences for turning metrics, we used linear mixed models. The linear mixed models included group, age, and sex as fixed effects, and subjects included as a random effect using unbounded variance components and the restricted maximum likelihood (REML) method. 

To assess differences for the region of interest (ROI) cortical thickness and TMS measures, again we used linear mixed models. The linear mixed model included group, hemisphere, age, and sex as fixed effects, group × hemisphere as an interaction, and subjects included as a random effect using unbounded variance components and the REML method. In all cases, inspection of residual plots showed equal variance. Post-hoc analyses were not performed as no interactions demonstrated significance.

Pearson’s correlation coefficients were used to assess correlations between hemisphere-specific cortical thickness measures and turning variables, and hemisphere-specific TMS metrics and turning variables. Correlations were corrected for multiple comparisons using the Bonferroni correction method. Bidirectional correlation strengths (positive or negative) were classified as very strong (±0.9–1.0), strong (±0.7–0.9), moderate (±0.5–0.69), weak (±0.3–0.49), and negligible (±<0.30) [49].

## 3. Results

### 3.1. Participants

We were unable to detect a reliable ‘hot spot’ in certain hemispheres in one control and four multiple sclerosis participants, although the usable data from those participants were maintained for all further analysis. The demographic and clinical characteristics are summarized in Table 1. No significant differences were observed for age (*F*(1, 48) = 0.13, *p* = 0.72) or sex (χ^2^(2) = 0.36, *p* = 0.55). Additionally, weight, height, and BMI were not significantly different between groups. 

### 3.2. Turning Performance

In-place 360° fast pace turns demonstrated significant differences between groups for turn duration (*F*(1, 45) = 20.50, *p* < 0.001), peak turn velocity (*F*(1, 45) = 23.82, *p* < 0.001), and turn angle (*F*(1, 45) = 4.20, *p*= 0.046). Specifically, PwMS demonstrated significantly longer turn durations, and slower peak turn velocities, and turn angles closer to 360°.

Continuous, but alternating, self-selected pace 360° in-place turns demonstrated significant differences for all turn variables. PwMS demonstrated significantly longer turn durations (*F*(1, 45) = 6.70, *p* = 0.01), slower peak turn velocities (*F*(1, 45) = 6.90, *p* = 0.01), reduced (i.e., nearer to 360°) turn angles (*F*(1, 45) = 7.03, *p* = 0.01), and fewer total turns completed (*F*(1, 45) = 5.39, *p* = 0.03) compared to control participants.

Turns completed during the self-selected pace 2-min walk test did not demonstrate significant differences between groups for turn duration (*F*(1, 45) = 2.56, *p* = 0.12), peak turn velocity (*F*(1, 45) = 0.63, *p* = 0.43), turn angle (*F*(1, 45) = 0.10, *p* = 0.75), or number of steps to complete the turn (*F*(1, 45) = 3.71, *p* = 0.06). The data for each type of turn and variable collected are reported in Table 2.

### 3.3. Motor Cortex Thickness

Cortical thickness of the precentral gyrus demonstrated a significant main effect of group (*F*(1, 45) = 9.41, *p* = 0.004) with reduced thickness in PwMS, and a significant main effect of hemisphere (*F*(1, 47) = 9.10, *p* = 0.004) with the right hemisphere demonstrating less cortical thickness for both groups. No significant group × hemisphere interaction (*F*(1, 47) = 0.001, *p* = 0.98) was observed. The paracentral gyrus demonstrated a significant main effect of group (*F*(1, 45) = 10.86, *p* = 0.002) such that PwMS demonstrated reduced cortical thickness. Further, there were no effect of hemisphere (*F*(1, 47) = 1.23, *p* = 0.27) or group × hemisphere interaction (*F*(1, 47) = 0.52, *p* = 0.47). Thickness values for each group and hemisphere can be observed in Figure 4.

### 3.4. TMS

Maximal strength for the TA muscles demonstrated a significant main effect of group (*F*(1, 45) = 15.53, *p* < 0.001), with the multiple sclerosis cohort demonstrating overall weaker dorsiflexor output. 

No significant effects were found for RMT between groups (*F*(1, 45) = 0.09, *p* = 0.33), hemispheres (*F*(1, 47) = 0.25, *p* = 0.61), or the group × hemisphere interaction (*F*(1, 47) = 0.12, *p* = 0.73). Motor cortex excitability was measured via MEP amplitude relative to the pre-stimulation mean muscle activity. For MEP amplitude normalized to the pre-stimulation average, there were no main effects of group (*F*(1, 42.4) = 0.02, *p* = 0.90), hemisphere (*F*(1, 43.1) = 1.57, *p* = 0.21), or a group × hemisphere interaction (*F*(1, 43.1) = 2.45, *p* = 0.12).

For the inhibitory TMS measures, no significant effect of group (*F*(1, 43.9) = 0.05, *p* = 0.83), hemisphere (*F*(1, 44.4) = 1.51, *p* = 0.23), or interaction (*F*(1, 44.4) = 0.66, *p* = 0.42) was found for the cortical silent period (cSP) duration. For %dSP_AVE_ there was a significant main effect of group (*F*(1, 45.6) = 6.54, *p* = 0.01), although no main effect of hemisphere (*F*(1, 45.9) = 0.18, *p* = 0.67) nor a group × hemisphere interaction (*F*(1, 45.9) = 1.02, *p* = 0.32). The results show significantly reduced %dSP_AVE_ for PwMS compared to their neurotypical counterparts. For %dSP_MAX_ a significant main effect of group (*F*(1, 44.1) = 5.87, *p* = 0.02) was observed, such that PwMS demonstrated reduced %dSP_MAX_. However, no main effect of hemisphere (*F*(1, 44.3) = 0.23, *p* = 0.23) or group × hemisphere interaction (*F*(1, 44.2) = 0.15, *p* = 0.70) was revealed. TMS-related measures for each group and hemisphere are reported in Table 3.

### 3.5. Associations 

#### 3.5.1. Associations between 360° In-Place Fast Turns and Neurophysiology and Neuroanatomical Structure

Controls did not demonstrate any significant associations between turn variables and the TMS or MRI measures. In contrast, PwMS demonstrated significant negative associations between turn duration and left hemisphere inhibitory measures. Additionally, a significant negative association was revealed between turn duration and paracentral gyri thickness, together indicating that those with greater levels of left hemisphere corticospinal inhibition and paracentral thickness demonstrated shorter turn durations. For the right hemisphere, a significant positive association was observed between turn angle and MEP amplitude, indicating that those with greater right hemisphere excitability demonstrated greater 360° turn angles. Figure 5 details the associations between 360° turn measures and hemisphere-specific TMS and MRI variables.

#### 3.5.2. Associations between 360° Self-Selected Pace In-Place 1-min Continuous Turns and Neurophysiology and Neuroanatomical Structure

For the 1 min of continuous 360° normal pace turns, controls did not demonstrate any significant associations between turning variables and the TMS or MRI measures. Alternatively, PwMS demonstrated significant associations between turn duration and number of turns completed and left hemisphere %dSP_AVE_ and %dSP_MAX_. These associations demonstrated that those individuals with greater levels of inhibitory capacity performed turns in less time and completed more total turns over the course of 1 min. Additionally, significant associations were observed between turn velocity and left hemisphere paracentral and precentral cortical thickness, such that those with greater thickness demonstrated faster turn velocities. Specific to the right hemisphere, only one significant association showed a positive association between turn angle and MEP amplitude, indicating that those with greater right hemisphere MEP amplitude demonstrated greater 360° turn angles (i.e., further from 360°). Figure 6 details associations between 360° turn measures and hemisphere-specific TMS and MRI variables.

#### 3.5.3. Associations between 180° Self-Selected Pace Turns While Walking and Neurophysiology and Neuroanatomical Structure

For the 180° self-selected pace turns, neurotypical controls demonstrated a significant negative association between turn duration and right hemisphere precentral gyri thickness, indicating that neurotypical controls with greater precentral thickness perform turns in less time. No other significant associations were observed between turn variables and hemispheric TMS or MRI measures for neurotypical controls. PwMS demonstrated significant positive associations between turn velocity and left hemisphere silent period duration and %dcSP_AVE_. These associations indicate that those with greater levels of inhibition demonstrate faster turn velocities. Moreover, a significant association was observed between turn duration and right hemisphere MEP amplitude, indicating that those with greater excitability perform turns in less time. Figure 7 details associations between 180° turn measures and hemisphere-specific TMS and MRI variables. Refer to Appendix A, which provides correlation scatter plots for all left hemisphere significant associations between neuroanatomical structure and turning performance and neurophysiological function and turning performance.

## 4. Discussion

### 4.1. Turning

Significant differences were observed for each 360° turn measure, indicating that PwMS, independent of speed, maintain reduced turning performance compared to controls. These current results closely align with prior studies reporting increased turn duration and velocity in PwMS [50,51,52,53]. Turn angles for the 360° turns were closer to the instructed 360° mark in PwMS compared to controls. This finding coincides with Shah et al., who reported reduced turn angles in PwMS during 7 days of continuous mobility monitoring when compared to controls [54]. While the study designs were different, we postulate that PwMS perform reduced turn angles due to an abundance of caution and the ability to produce an appropriate compensatory response if needed [55]. However, the true functional significance of reduced turn angles in PwMS remains unknown. 

No significant differences were observed between groups for any of the 180° turn variables. These results are partially divergent to previously reported 180° turns differences between PwMS and controls. For instance, Spain et al. reported similar peak turn velocities between groups during an instrumented Timed Up and Go (iTUG) test but significantly longer 180° turn durations in PwMS [52]. We suspect the differences are likely task related given that the iTUG and the 2-min walk have unique task objectives. Together, these results may indicate that PwMS demonstrate altered turning characteristics and subsequently reduced turning performance for 360° in-place turns although no differences for 180° turns while walking. Possibly indicating that task complexity and turn style may provide important turn related kinematic differences between PwMS and controls.

### 4.2. MRI

The precentral and paracentral gyri were chosen a-priori based on the known influence and integration of motor commands converging within those cortical regions along with prior evidence demonstrating disease-related susceptibility to cortical atrophy [14,56]. The present results are consistent with prior investigations demonstrating motor cortex gray matter thinning in PwMS with the RRMS phenotype [14,56,57,58,59,60,61].

### 4.3. TMS

A necessary measure for any TMS study is the determination of the RMT, thereby normalizing responses across study participants. The majority of studies assessing RMT have reported no differences between healthy controls and PwMS, though greater thresholds for PwMS have been reported (see review by Snow et al. [62]). In alignment, our results demonstrated no significant differences between groups for RMT [62]. While the mechanisms associated with RMT and the generation of transmembrane excitation remain inconclusive, these results may indicate that both groups demonstrated similar pyramidal fiber orientation within the simulated region. Indeed, modelling studies indicate that pyramidal cell structure and orientation influence the electrical fields produced by TMS, which are necessary for MEP production [63,64,65,66].

Glutamatergic activity is a commonly measured neurophysiological outcome in the multiple sclerosis literature and often assessed via the MEP amplitude [62]. With no observed differences between groups, the current results are consistent with numerous studies that also accounted for important covariates [62]. These results may indicate that our two cohorts demonstrated similar density of cortico–motor neuronal projections stemming from the motor cortex, thus providing similar glutamatergic responses to TMS [67]. However, it should be noted that MEP measures and their physiological underpinnings are complex and remain not fully understood [47].

Inhibitory neurotransmission has demonstrated associations with motor learning, neural plasticity, and motor control [31,68]. However, the role of inhibitory neurotransmission within the corticospinal system of PwMS remains inconclusive, with studies reporting inconsistent findings [60,69,70,71,72,73,74]. Moreover, there remains a substantial lack of TMS-related studies focused on assessing the lower limbs. Indeed, only one other study, performed by Tataroglu et al., has explored inhibitory activity via single pulse TMS targeting the lower limbs in PwMS [73]. Their results demonstrated significantly longer cSP durations for PwMS compared to neurotypical controls and, furthermore, greater durations for those with progressive phenotypes of the disease. In contrast, the current results demonstrated no differences in silent period duration between PwMS and controls. Upon further exploration, Tataroglu et al. reported an average silent period duration of 118.1 ± 74 ms for their RRMS group, while the current multiple sclerosis cohort demonstrated similar durations being 108.3 ± 50 ms and 123.9 ± 68 ms for the left and right hemispheres, respectively. Interestingly, and dissimilar to our results, considerably shorter silent period durations were reported for their control cohort [73]. A variety of factors can influence silent period duration including stimulation intensity, contraction force, and number of stimulations, among others [33,46,75,76]. These protocol differences could contribute to the differences observed, limiting the comparative interpretation between results.

Intrahemispheric and interhemispheric inhibition are most often quantified through the silent period duration; however, percent average and max depth of the silent period have been used to quantify levels of interhemispheric inhibitory activity [48,77]. Studies assessing interhemispheric inhibition have shown %dSP_AVE/MAX_ demonstrate greater sensitivity for delineating between young and older adults compared to the conventionally reported silent period duration [46,78,79], although these measures have not been widely reported for intrahemispheric (i.e., cSP) inhibitory analyses. One potential reason could be due to the greater inhibitory influence (i.e., EMG amplitude during the silent period nears 100% muscle deactivation) for cSPs compared to iSPs, meaning researchers may feel these measures would be less likely to demonstrate group differences [46]. While the physiological mechanisms underlying these metrics remain under investigation, they are suspected to provide unique perspectives of GABAergic inhibitory influence and have been characterized as both sensitive and reliable [48,80,81,82,83] were a greater percent indicates larger corticospinal inhibitory influence at the muscle [46]. Our results demonstrated similar silent period durations, albeit with significantly reduced inhibitory influence in PwMS compared to controls. While these results are a new expression of inhibitory data, they align with prior studies reporting no differences or reduced inhibitory activity in PwMS compared to neurotypical adults [69,70,71,72]. The current inhibitory results add to the existing literature by revealing temporally similar silent period durations to the lower limbs between groups while demonstrating reduced inhibitory influence on the TA in PwMS. 

### 4.4. Associations

Given that the data did not demonstrate a hemispheric difference for inhibition, we could have averaged the hemispheric results; however, we decided to maintain independent hemispheric values. Principally, this decision was guided by evidence of brain lateralization, or the notion of each hemisphere integrating specific responses and actions [84]. For instance, Fling et al. demonstrated associations between proprioceptive balance control and microstructural integrity of Brodmann Area 3a to be restricted entirely to the right hemisphere in PwMS [84]. Moreover, voluntary inhibition of manual movements (via go–no-go tasks) has been postulated to be right-lateralized to the frontal-basal ganglio–thalamic pathway [85], although evidence to contradict that assumption exists, suggesting that both hemispheres work in cooperation [86]. Furthermore, it has been postulated that the left hemisphere may be responsible for planning motor actions, and subsequently has become specialized in regulating well-established patterns of behavior characterized as routine, familiar, and internally directed [86,87]. While the contributions for each hemisphere remain under investigation, particularly for lower limb directed movements, evidence indicates each hemisphere has some degree of functional independence. 

#### 4.4.1. Neuroanatomical Associations

Gray matter atrophy has been associated with clinical indicators of physical disability, cognitive decline, and disease duration [12,14,88,89,90,91]. Importantly, gray matter atrophy demonstrates independence from white matter degradation and provides stronger associations with clinical indicators [12,13,88,91,92]. While gray matter atrophy demonstrates clinically relevant correlations in multiple sclerosis, recent reports suggest atrophic patterns are predominately regional rather than globally diffuse [56,61]. Furthermore, regional patterns of atrophy appear to provide distinct relationships with functional disability [56,61]. Specifically, it has been observed that left-lateralized atrophic clusters of the sensorimotor cortex demonstrate significant negative associations with disease severity using the Expanded Disability Status Scale (EDSS) [56,93]. Given that the EDSS has a bias towards locomotor disability, this negative association suggests the atrophy of sensorimotor region influences functional disability. Consistent with prior results demonstrating an association between the disease severity and cortical thickness of the motor cortex, our results further demonstrate motor cortex atrophy associated with specific characteristics of turning performance, revealing that reduced cortical thickness of the precentral and paracentral gyri correlates with 360° turn duration, peak velocity, and angle in PwMS and 180° turn duration in controls. Therefore, providing further evidence that motor cortex atrophy influences functional disability, specifically turning performance. 

To our knowledge, one other study performed by Lorefice and colleagues assessed the associations between global cortical atrophy and components of the iTUG [94]. The authors showed a significant negative association between global cortical gray matter volume and completion time along with significant positive associations between turn velocity and global cortical gray matter volume [94], thus, revealing a relationship between cortical gray matter structure and lower limb dynamic motor performance, where greater cortical gray matter volume relates to faster turns and greater iTUG performance. It must be noted that turns performed during the iTUG are 180°, which in our study, only turn duration for the control group demonstrated significance. We believe these differences are a product of unique task objectives and associating specific turning variables to global, rather than regional, measures of atrophy. 

#### 4.4.2. Neurophysiologic Associations

Glutamate and GABA are critical for the development and regulation of descending motor commands. While intrahemispheric glutamatergic activity describes the balance between excitation and inhibition, TMS-related glutamatergic measures are often less sensitive to motor control assessments compared to inhibitory neurotransmission [31,95]. The current results show a similar trend, revealing many more significant inhibitory associations compared to excitatory associations with turning performance. While our results revealed no group differences for normalized MEP amplitude, two associations emerged as significant, with smaller amplitudes correlating with reduced turning performance in PwMS. Corresponding with our results, prior research has shown weak-to-moderate associations between disease severity and MEP amplitude, such that reduced glutamatergic activity relates to greater disease severity [25,62].

While older adults and PwMS are distinct in many ways, they do demonstrate similar mobility deficits, for instance, both groups demonstrate very similar turning characteristics [96]. Additionally, the associations from the present study demonstrate similarities to our previously published healthy aging data, such that greater inhibition in the impaired groups (e.g., PwMS and older adults) relates to better turning performance. While cSP duration did not demonstrate a groupwise difference, it did reveal a negative association with 360° in-pace fast turn duration and a positive association with 180° peak turn velocity. These results may indicate that a temporal inhibitory influence (i.e., cSP duration) on lower limb muscles is important for temporally mediated turning movements. Interestingly, the two measures of inhibitory influence demonstrated associations with additional turning variables, potentially implying that inhibitory influence could be a sensitive measure for lower limb dynamic movements requiring higher level neural command.

While there were many associations pertaining to neurophysiological activity and turning performance for PwMS, no significant association was observed in the control group. However, directionally, the associations in the control group broadly opposed those in the multiple sclerosis group, such that neurotypical adults with greater inhibitory activity demonstrated reduced turning performance. Interestingly, this result is like previously published observations for the inhibitory control of turning in healthy young adults [33]. While these results demonstrated a similar pattern to young adults, it must be noted that the inhibitory adaptations associated with middle-aged adults largely remains unexplored. However, we postulate that healthy middle-aged adults may rely on alternate neural resources such as subcortical and/or spinal level modulation for successful bilateral lower limb control [97,98].

Interestingly, all the correlations between turning and inhibition were lateralized to the left hemisphere. These findings are particularly interesting and demonstrate novel findings regarding inhibitory lateralization and turning performance in PwMS. While the significance of left hemisphere lateralization and turning performance in PwMS remains unknown, left hemispheric specialization is thought to play a significant role in enhancing movement planning and execution [87]. Further, studies have provided evidence suggesting a particular role for the left hemisphere in motor learning tasks that require movement planning and execution for future actions. These ideas are consistent with pathological populations that have demonstrated greater left, rather than right, hemispheric damage resulting in ideomotor apraxia, a disorder where the observed spatiotemporal motor deficits are thought to arise from impaired planning due to damage of the left frontal and parietal brain regions [87,99]. Additionally, the left parietal cortex has been implicated for its role in the preparation of selected overt movements and the decision of which limb to use for particular tasks [100,101]. Collectively, it appears that the left hemisphere is particularly well suited for movement planning and execution, which has been shown to be disrupted in PwMS [102]. These results may indicate a compensatory inhibitory lateralization meant to assist in properly executing the planned turn. Interestingly, the control group did not demonstrate similar associations, indicating the utilization of different neural mechanisms to perform the same task, though the broad age range of the control cohort may have diminished any associations.

To date, no studies have assessed both cortical thickness and neurophysiological activity in PwMS specific to the lower limbs. Moreover, no studies have incorporated objective measures of dynamic lower limb movements to identify neural mechanisms associated with performance. This study provides evidence to suggest a relationship between neuroanatomical structure and neurophysiological function of the motor cortex and turning performance in PwMS.

### 4.5. Limitations

The inclusion of only individuals with RRMS with an average EDSS of 3.5 could be a limitation, as it does not account for other disease phenotypes or greater levels of disease severity. Therefore, these results should be interpreted responsibly when relating them to other phenotypes or severity levels. Given that the leg region of the motor cortex (i.e., paracentral gyrus) lies within the longitudinal fissure, there was a slight possibility of TMS stimulation over-flow to the homologous paracentral gyrus. However, during ‘hot spot’ detection, MEPs were collected simultaneously from both legs with special care taken to elicit MEPs from the targeted hemisphere and corresponding TA. Additionally, we did not include specific MRI sequences suitable for the detection of gray matter lesions, such as a double inversion recovery sequence [103]. However, the analysis performed for this particular study integrated T1 + T2-FLAIR data, which have been shown to significantly improve cortical segmentation and parcellation and the identification of cortical atrophy [44]. Despite the vertex-wise analysis, we cannot rule out that white matter atrophy could have influenced the results, while FreeSurfer segmentations were rigorously assessed for errors, differences could exist between subjects with or without sulci deformation (i.e., widening) due to white matter atrophy. However, it remains unclear whether this would preferentially affect the results of certain cortical regions. 

## 5. Conclusions

The results from this study demonstrate that in PwMS, neuroanatomical structure and neurophysiological function are related to turning performance. Upon closer inspection, the associations between inhibitory activity and turning performance are characteristically stronger associations than those between motor cortex thickness and turning performance. While the statistical assumptions were not met, a mediation analysis could provide greater clarity as to the degree of influence inhibition has on the association between cortical thickness and turning performance. From these results, PwMS perform turns more similar to their control counterparts when greater inhibitory activity and motor cortex thickness are present. Finally, these results indicate that PwMS may utilize higher order cortically controlled neural mechanisms to perform dynamic movements typically associated with fall risk.

## Figures and Tables

**Figure 1 sensors-23-07629-f001:**
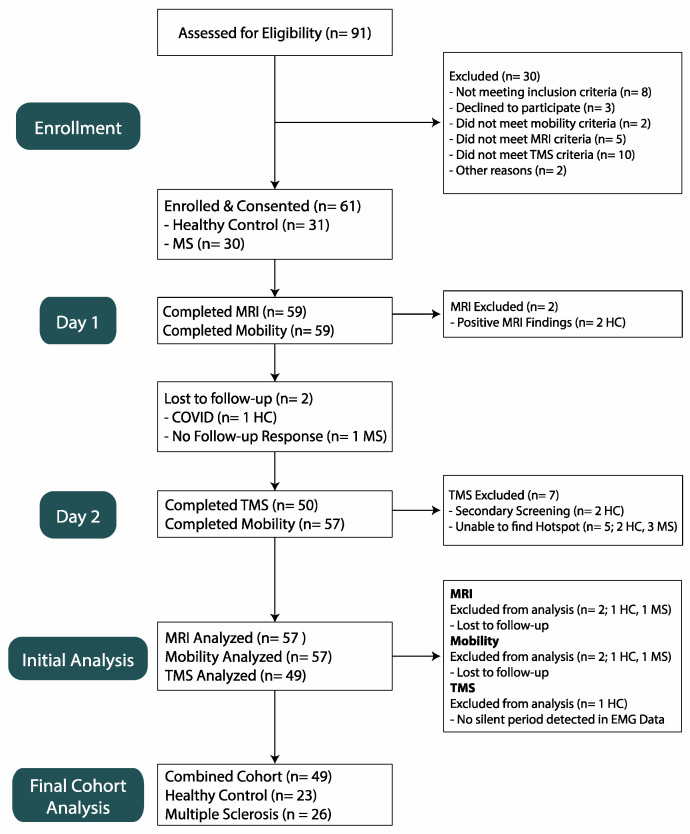
Flow chart of study enrollment.

**Figure 2 sensors-23-07629-f002:**
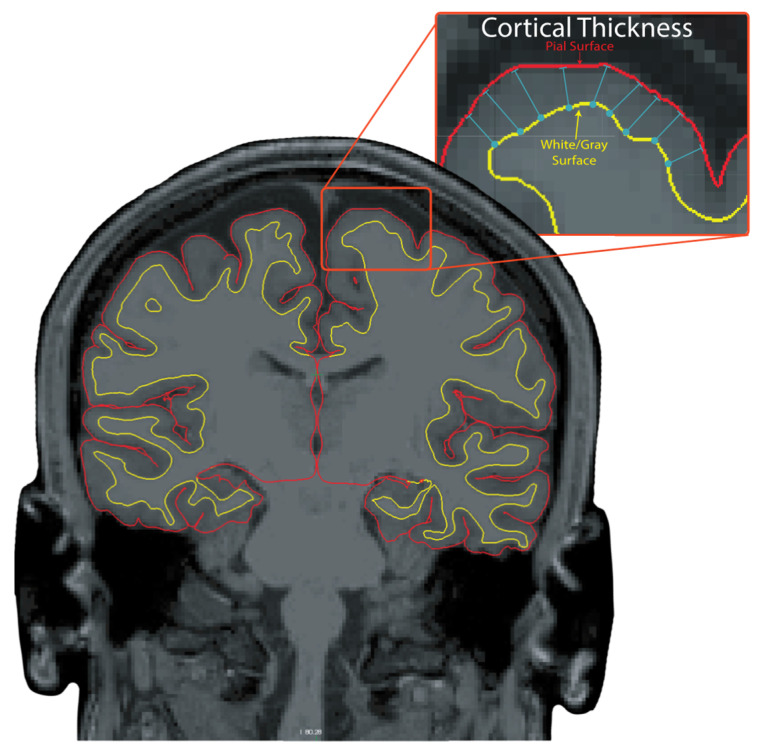
Visual description of how cortical thickness is measured between the white/gray matter surface and the pial surface in FreeSurfer.

**Figure 3 sensors-23-07629-f003:**
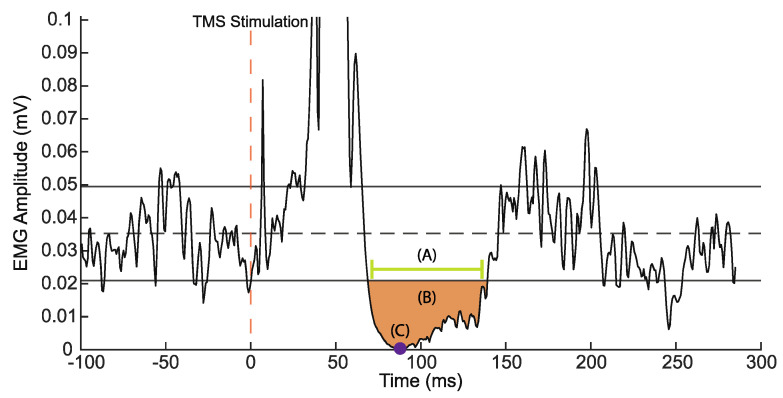
Representative silent period metrics and silent period duration calculations. (A) Silent period duration (ms), (B) %dSP_AVE_, (C) %dSP_MAX_.

**Figure 4 sensors-23-07629-f004:**
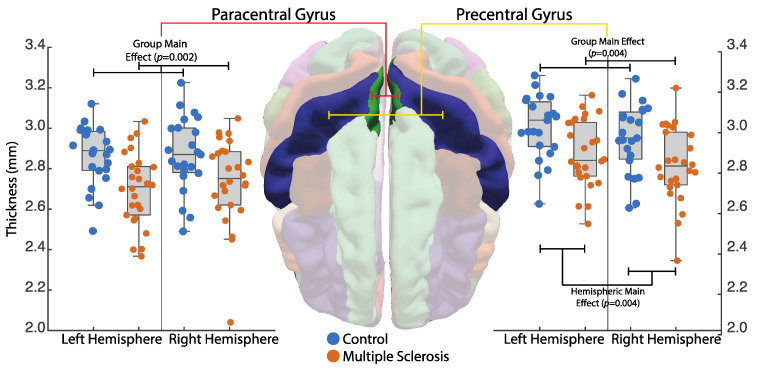
Hemispheric ROI cortical thickness differences between controls and PwMS.

**Figure 5 sensors-23-07629-f005:**
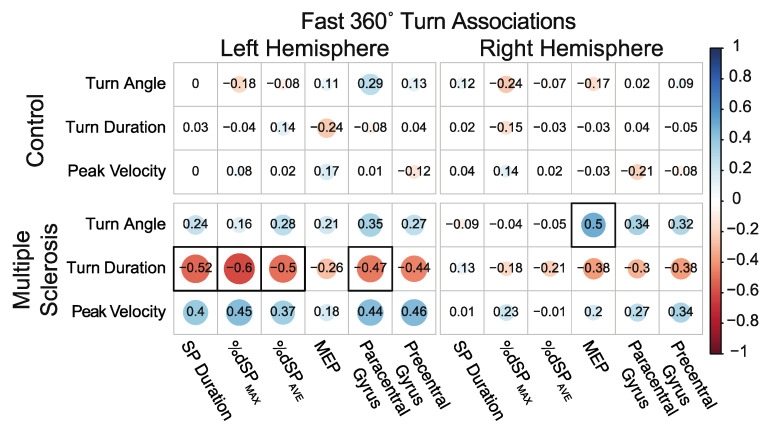
Associations between 360° in-place fast turns and neurophysiology and neuroanatomical structure. Size, color, and opacity of circles indicate the direction and strength of association. Boxes with a thick black border indicate statistical significance after correcting for multiple comparisons.

**Figure 6 sensors-23-07629-f006:**
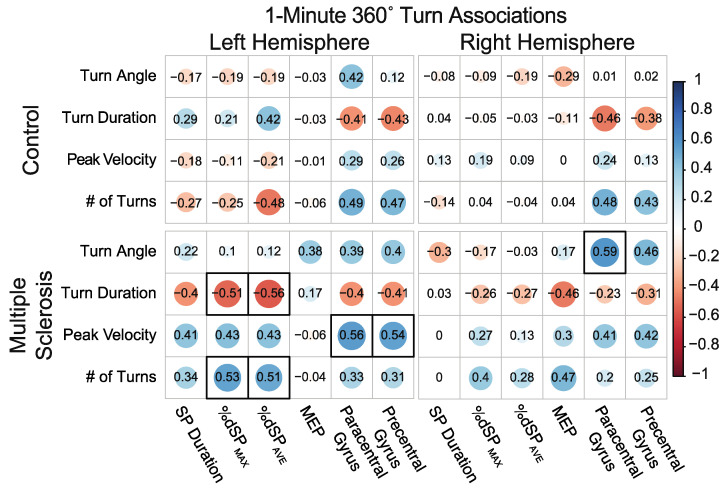
Associations between 360° self-selected pace in-place 1-min continuous turns and neurophysiology and neuroanatomical structure. Size, color, and opacity of circles indicate the direction and strength of association. Boxes with a thick black border indicate statistical significance after correcting for multiple comparisons.

**Figure 7 sensors-23-07629-f007:**
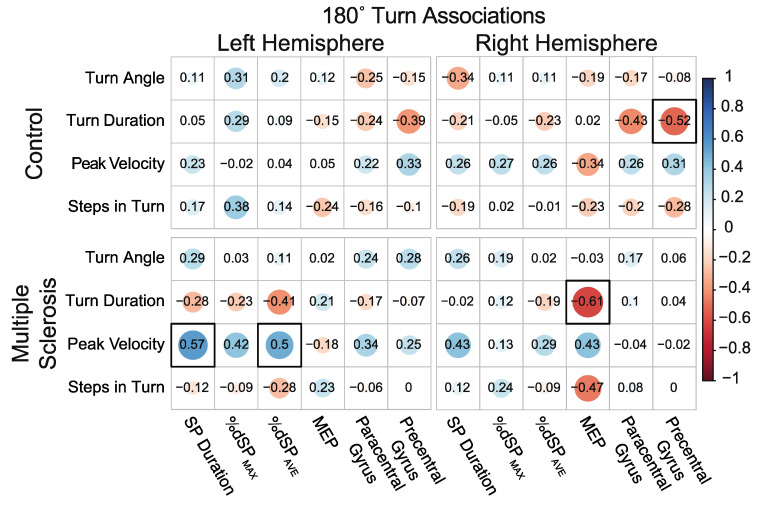
Associations between 180° self-selected pace turns while walking and neurophysiology and neuroanatomical structure. Size, color, and opacity of circles indicate the direction and strength of association. Boxes with a thick black border indicate statistical significance after correcting for multiple comparisons.

**Table 1 sensors-23-07629-t001:** Participant characteristics, demographics, and selected clinical data for each group.

Variable	Healthy Control	Multiple Sclerosis	*p*-Value
Sex, female/male	15/8	19/7	
Age (years), mean ± SD	46.76 ± 15.93	48.19 ± 11.95	0.72
Height (cm), mean ± SD	169.85 ± 7.59	165.73 ± 7.32	0.06
Weight (kg), mean ± SD	71.89 ± 13.20	68.39 ± 9.35	0.29
BMI, mean ± SD	24.79 ± 3.35	24.99 ± 3.84	0.84
Disease duration (years), mean ± SD	-	11.69 ± 10.72	
EDSS, median [range]	-	3.5 [0–4]	

BMI = body mass index; EDSS = Expanded Disability Status Scale.

**Table 2 sensors-23-07629-t002:** Between group differences adjusting for age and sex for each turn measure and variable.

Turn Variables	Healthy Control	Multiple Sclerosis	*p*-Value
360° Fast Turn			
Turn Duration (s), mean ± SD	1.85 ± 0.30	2.44 ± 0.69	<0.0001
Turn Velocity (°/s), mean ± SD	368.61 ± 65.98	293.30 ± 80.36	<0.0001
Turn Angle (°), mean ± SD	384.60 ± 11.72	378.57 ± 15.36	0.05
1 Min 360° Turns			
Turn Duration (s), mean ± SD	2.87 ± 0.46	3.36 ± 0.90	0.01
Turn Velocity (°/s), mean ± SD	232.42 ± 38.43	203.21 ± 50.25	0.01
Turn Angle (°), mean ± SD	391.05 ± 15.25	377.53 ± 25.77	0.01
Turns Completed (#), mean ± SD	19.43 ± 3.26	16.96 ± 4.40	0.02
180° Turns			
Turn Duration (s), mean ± SD	2.04 ± 0.31	2.18 ± 0.37	0.12
Turn Velocity (°/s), mean ± SD	212.14 ± 29.38	206.16 ± 38.40	0.43
Turn Angle (°), mean ± SD	184.44 ± 7.93	183.48 ± 7.91	0.75
Steps in Turn (#), mean ± SD	3.57 ± 0.67	3.96 ± 0.67	0.06

**Table 3 sensors-23-07629-t003:** Average TMS-related measures separated by group and hemisphere.

TMS Variable	Hemisphere	Healthy Control	Multiple Sclerosis
Resting Motor Threshold (%MSO), mean ± SD	Left	68.96 ± 8.09	66.77 ± 7.42
	Right	69.09 ± 7.68	67.50 ± 7.91
MEP Amp (mV), mean ± SD	Left	16.89 ± 7.38	18.41 ± 8.58
	Right	17.16 ± 6.39	14.91 ± 6.27
Silent Period Duration (ms), mean ± SD	Left	115.66 ± 46.24	108.26 ± 50.04
	Right	117.83 ± 42.54	123.94 ± 67.88
dSP_AVE_ (%), mean ± SD	Left	76.89 ± 7.27	70.29 ± 9.63
	Right	75.13 ± 7.81	71.09 ± 7.84
dSP_MAX_ (%), mean ± SD	Left	95.98 ± 2.87	92.89 ± 5.32
	Right	95.35 ± 3.10	93.02 ± 4.55

%MSO = percent maximal stimulator output; dSP_AVE_ (%) = average percent depth of silent period; dSP_MAX_ (%) = maximal percent depth of silent period.

## Data Availability

The data presented in this study are available on request from the corresponding author.

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
