# Peer review of "Links between Neuroanatomy and Neurophysiology with Turning Performance in People with Multiple Sclerosis"

_sensors, 2023, doi:10.3390/s23177629_

Round 1

Reviewer 1 Report

Congratulations to the researchers. Acceptable for publication if appropriate for the editor and other reviewers

Congratulations to the researchers. Acceptable for publication if appropriate for the editor and other reviewers

Author Response

We would like to thank Reviewer 1 for their review of our manuscript.

Reviewer 2 Report

The paper is relevant and represents an example of a real scientific work.

The Authors convincingly proved that both cortical thickness and inhibitory activity were associated with turning performance in people with multiple sclerosis.

In comparison to the works in refs. 33, 40 and 50, 51, 58 the authors of the study took a further step in monitoring the turning ability, especially in patients with MS.

The undertaken hypotheses and the methodology applied in this study are relevant.

The population of MS participants examined twice is good enough for the reliable statistical proceeding, the analyses of the results, and the drawing of the proposed proper conclusions.

Minor revisions should be considered:

The Abstract is very concise which is good, but lacks at least the most important statistical data. The number of participants with MS and EMG recordings from TA should be mentioned.

The Conclusion (lines 23-24) should be extended because it resembles in fact the statements from the previous studies, omitting the novelty of contemporary presented results. Do authors consider to conclude about the clinical relevance of their observations?

M&M section. 

The right and lower parts of the flow chart seem to be cut.

Exclusion criteria for TMS should be listed.

The EMG and MEP recording parameters (amplification and time base) should be mentioned.

The content of the sentence in lines 185-187 better fits to the beginning of the Results section.

Results

The abbreviation of Table 1  (line 215) should be supplemented with ….”and selected clinical data”… .

The results are clearly and precisely described. The authors might consider the presentation of the TMS-evoked MEP recordings in patients and healthy controls for comparison.

Discussion

Perfect, but I cordially invite the authors for some words of the possible clinical application of their results.

The selection of references is outstanding to the issues discussed in the paper.

The style of the MDPI citation needs a slight improvement, in general. 

Authors sometimes use short journal names, sometimes not.

Ref. 18 needs completion.

Author Response

We would like to thank Reviewer 2 for their thoughtful review of our manuscript. Please see responses to comments below.

Abstract:

The Abstract is very concise which is good, but lacks at least the most important statistical data. The number of participants with MS and EMG recordings from TA should be mentioned.

Thank you for the comment, for brevity, we refrained from adding statistical values within the abstract. We however agree with the reviewer that important details were lacking. As such, we have adjusted the abstract accordingly.

The Conclusion (lines 23-24) should be extended because it resembles in fact the statements from the previous studies, omitting the novelty of contemporary presented results. Do authors consider to conclude about the clinical relevance of their observations?

Thank you for the comment. Pertaining to potential clinical relevance we have offered further interpretation of the results.

M&M section:

The right and lower parts of the flow chart seem to be cut.

Thank you for noticing this error. We have fixed and uploaded a new version of the Figure 1.

Exclusion criteria for TMS should be listed.

Thank you for the comment. We have expanded the study exclusion criteria with particular emphasis on MRI and TMS.

The EMG and MEP recording parameters (amplification and time base) should be mentioned.

The EMG collection settings are mentioned in section 2.8 TMS Processing and read as follows: “EMG data was collected at 2,000 Hz (BIOPAC Systems, Inc., Santa Barbara, CA). Offline, EMG data was filtered using a combination bandpass filter (5 – 500Hz) with a 60Hz Nyquist filter through AcqKnowledge software.” If there is other information Reviewer 2 would like us to add, we kindly request additional details.

The content of the sentence in lines 185-187 better fits to the beginning of the Results section.

We agree and have moved this sentence to the beginning of the results section.

Results:

The abbreviation of Table 1  (line 215) should be supplemented with ….”and selected clinical data”… .

Thank you, we have added this recommendation to the title of Table 1.

The results are clearly and precisely described. The authors might consider the presentation of the TMS-evoked MEP recordings in patients and healthy controls for comparison.

Thank you for the comment. Since the MEP amplitude results did not demonstrate a significant effect of group or hemisphere, or even an interaction we felt an additional figure would not significant context to readers.

Discussion:

Perfect, but I cordially invite the authors for some words of the possible clinical application of their results.

We greatly appreciate the feedback and have added the following sentence to the conclusion. “Finally, these results indicate that PwMS may utilize higher order cortically controlled neural mechanisms to perform dynamic movements typically associated with fall risk.” Given the observational nature of this study and because neither author is a clinician, we would like to stay within the confines of our interpretation and expertise.

The selection of references is outstanding to the issues discussed in the paper. The style of the MDPI citation needs a slight improvement, in general. Authors sometimes use short journal names, sometimes not. Ref. 18 needs completion.

Thank you for carefully reviewing the manuscript in its entirety. We downloaded the MDPI citation manager template and updated the references accordingly the their specifications.

Reviewer 3 Report

This reviewer was very impressed with this manuscript which showed significant correlations between neurophysiology and neuroanatomy (cortical thickness) in patients with multiple sclerosis.  Here are some suggestions to help improve the manuscript.

1) Show an example high res T1 MR scan showing the coritcal thickness measurement.

2) For a few of the most significant correlations, show a scatter plot with cortical thickness value on the x axis and  behavioral score on the y-axis.  Show the regression line and correlation coefficient.

Author Response

This reviewer was very impressed with this manuscript which showed significant correlations between neurophysiology and neuroanatomy (cortical thickness) in patients with multiple sclerosis. Here are some suggestions to help improve the manuscript.

We appreciate the comments and thoughtful recommendations provided by Reviewer 3.

Show an example high res T1 MR scan showing the cortical thickness measurement.

We have added a figure portraying a high resolution T1 that also visually depicts how cortical thickness is measured using FreeSurfer.

For a few of the most significant correlations, show a scatter plot with cortical thickness value on the x axis and behavioral score on the y-axis.  Show the regression line and correlation coefficient.

We thank the reviewer for the suggestion. We have put together a supplemental figure highlighting significant associations from the left hemisphere.